# Real-Time Mine Road Boundary Detection and Tracking for Autonomous Truck

**DOI:** 10.3390/s20041121

**Published:** 2020-02-18

**Authors:** Xiaowei Lu, Yunfeng Ai, Bin Tian

**Affiliations:** 1Waytous Inc., Beijing 100083, China; luxiaowei17@mails.ucas.edu.cn (X.L.); aiyunfeng@ucas.ac.cn (Y.A.); 2School of Artificial Intelligence, University of Chinese Academy of Sciences, Beijing 100190, China; 3State Key Laboratory of Management and Control for Complex Systems, Institute of Automation, Chinese Academy of Sciences, Beijing 100190, China

**Keywords:** unstructured road, road boundary detection, automatic truck, mine scene

## Abstract

Road boundary detection is an important part of the perception of the autonomous driving. It is difficult to detect road boundaries of unstructured roads because there are no curbs. There are no clear boundaries on mine roads to distinguish areas within the road boundary line and areas outside the road boundary line. This paper proposes a real-time road boundary detection and tracking method by a 3D-LIDAR sensor. The road boundary points are extracted from the detected elevated point clouds above the ground point cloud according to the spatial distance characteristics and the angular features. Road tracking is to predict and update the boundary point information in real-time, in order to prevent false and missed detection. The experimental verification of mine road data shows the accuracy and robustness of the proposed algorithm.

## 1. Introduction

Many special scenarios require the practical application of autonomous driving including industrial automation, construction, and mining [1]. Due to the closed nature of a mine, it is easier to achieve automatic driving in mine scenes. On the one hand, this high-risk environment is dangerous for the staff and the demand for unmanned vehicles is urgent. On the other hand, the mine scene has a single mechanical operation and the uncontrollable factors of the road condition are low. To ensure the safety of personnel and the efficiency of mine operations, the automation technology of the mine environment is developing rapidly. Environmental perception is a fundamental issue for the autonomous truck.

Existing technologies use different types of sensors for road detection: LiDAR sensors [2], radar sensors [3], and stereo vision camera [4]. Compared with the visual camera, the advantage of radar is that it can accurately obtain the position information of the object while obtaining a larger field of view. The most important advantage is that LiDAR is not affected by lighting and can be used throughout the day. LiDAR has a large amount of information, high dimension, and high precision, which is more suitable for the detection of mine roads. Many existing methods of road detection are based on visual sensors, and the result of the detection is the category (road or background) label of each pixel. Image-based road detection algorithms are limited by the diversity of road scenes and imaging conditions. Algorithms for road detection based on point cloud data rely more on algorithm rules than datasets. Wang et al. [5] proposed a double layer beam model method to effectively detect intersection roads. This method [6] effectively solves problems such as road discontinuities, obstructions, and corners. Due to the simple structure of the urban road, the above method is applicable to urban scenes and not to mine roads. As shown in Figure 1, the overall environment of the mine presents a spiral shape that leads to the particularity of the mine road. The particular challenges in road mines are as follows:There is a small slope on the road surface of the mine, as shown in Figure 2a. It is difficult to find which are the ground points and which are the elevated points by the height information of the point cloud.There is no boundary between the road surface and the non-road surface in the mine road, as shown in Figure 2b. The road boundary is determined based on the analysis and judgment of obstacles on the roadside. However, the location of obstacles on the roadside of the mine is irregular and discontinuous. Therefore, many of the extracted road candidate points are invalid.Compared with ordinary rural roads and suburban roads, there are potholes and convex hulls on the road surface of mine roads. Figure 2c shows the raw data of the mine point cloud and the status of the scanning line. The LiDAR scanning line is broken up by the clod or interrupted by the mound, thus it is difficult to extract the road boundary points.

To solve the above challenging problems in the mine environment, this paper proposes a robust and effective method to detect the boundary line of mine roads. Figure 3 shows the main content of the mine road boundary detection and tracking method. We obtain elevated points above the ground by detecting the ground of the mine road. Due to the particularity of the mine road, we propose a method of double meshing. This method is not only suitable for the uneven road surface of the mine, but also for the slope of the mine road. The road boundary candidate points are extracted at the elevated point, and the false points outside the road are filtered through a series of processes. We optimize and stabilize the results of road detection through road boundary point tracking. The main contributions of this paper are as follows:We propose a method that can effectively detect the boundaries of mine roads.We design a road boundary point extraction method to filter false points that are outside the road.We propose a road boundary point tracking method based on Kalman filter, which can improve the stability and accuracy of detection results.We built a dataset for a mine scene. We collected point cloud data for multiple different types of roads. The calibration of the ground and road boundary points was performed for each frame of data.

The rest of the structure in this paper is as follows. In Section 2, we review the related technologies for road detection. In Section 3, the method of road boundary point extraction and fitting is described in detail. In Section 4, the Kalman filter that ensures the stability of road boundary detection results and tracks road boundary locations is presented. The experimental results are presented in Section 5. The paper is concluded in Section 6.

## 2. Related Work

We briefly review the following two aspects for road detection based on point cloud data. The two most important aspects of road detection are the extraction of road boundary points and the fitting of the boundary points.

### 2.1. Road Boundary Points Extraction

Structured urban roads have highly consistent curbs and the road surface is flat. Unstructured road detection is more difficult. By referring to the relevant literature, the method of extracting road boundary points based on point cloud is roughly divided into three types: according to the height change and angle difference of the road boundary points in each scan line of the LiDAR [7,8,9,10]; extracting the boundary points in the area divided by elevated points after removing the ground information [11,12,13,14,15]; and extracting significant points based on road surface features after dividing the road from MLS (Mobile Laser Scanning) point clouds [16,17,18,19].

When the vehicle is traveling on a structured flat road, the LiDAR scan line that is transmitted to the ground is a smooth straight line or curve. A scan line that encounters a raised object on either side of the road changes the angle of the scan line. Then, the road boundary points are extracted by the linear features of the road surface or the angular features of the cubs. Hu K et al. [7] extracted candidate points by curvature change and height difference of point cloud between ground and curb. Yao et al. [20] used the ground projection range, intensity data, and line segment features to extract the curb points for each scan line. Han J et al. [9] extracted road boundary points based on the resolution of the LiDAR and the distance of the point cloud on the scan line. This method is suitable for unstructured country roads. However, the mine road scan line is not continuous, and the scan line at the turn can only scan one road boundary.

The obstacles on both sides of the urban structured road are shoulders. The general obstacles on both sides of the unstructured road include roadside weeds, trees, guardrails, high piles, etc. The road boundary point is obtained according to the height difference between the road surface and the obstacle. After the point cloud data are divided by the grid, the boundary points are extracted according to the height and distance information in the grid area. The meshing can be done using a square grid in a Cartesian coordinate system or a ray projection model in a polar coordinate system [11,12]. It is better to divide into different fan-shaped areas by a fixed angle [13]. To solve the road conditions at crossings and intersections, the beam model [14,15] was introduced in the road detection scheme [14]. The beam width is expanded, and the beam length is adaptively divided. Then, the beam length is used as a feature to train using SVM (Support Vector Machine) to classify the ramp. After the effective detection of complex intersection roads, Wang X et al. [5] proposed the concept of the underlying and high-level beam models.

The MLS system [17] achieves high density and precise point clouds on large areas of the road, making it more suitable for road extraction. Wang et al. [16] proposed a method based on local normal saliency to obtain road boundary points by adaptively extracting salient points. The surface information of mine roads is complex. The scan line information of the mine road is intermittent and incomplete, thus it is difficult to extract the boundary points. Zai D et al. [17] used supervoxel generation and 3D road features to extract road boundary points. The method uses an alpha-shaped algorithm and energy minimization based on graph cuts to extract road boundary points. However, this method does not apply to sparse point cloud data.

### 2.2. Road Boundary Points Filtering

After the road boundary points are extracted, the points are classified as a set of points for each road boundary. Then, the road boundary line fitting is performed on each of the point cloud sets. Since most of the road detection is only for straight roads, the boundary point classification is directly based on the coordinate difference between the left and right road boundaries [16,17,21]. Pfeiffer and Franke [22] proposed a linear discriminant analysis based clustering method to divide the boundary points. The sliding beam method is proposed in [15] to obtain the road dividing line, and the boundary points are classified by the road dividing line.

Due to the irregularity of the road boundary, the candidate point is not necessarily the boundary point of the actual road. The directly extracted road candidate points may be outside the road; optimization is required to obtain smooth road boundary points. Road boundary point optimization is divided into two parts. One part is to filter the points away from the actual boundary line. The other part is to fit the model of the road boundary line and add the boundary points that have not been extracted according to the model.

Yang B et al. [19] adopted a series of methods to obtain smooth road boundary points: candidate point segmentation; outlier filtering; adjacent collinear segmentation merge; and B-spline algorithm, which increases boundary points that cannot be extracted on the boundary line. However, this method is more suitable for straight roads. Hata and Wolf [23] introduced a regression filter, but this filter will only have good results for obstacles in the road and cannot eliminate candidate points outside the road. Chen T. et al. [24] used some feature points as initial curb points to obtain boundary points using Gaussian Process Regression. Two consecutive one-dimensional Gaussian process regressions are utilized to construct a model of the left and right road boundaries.

## 3. Road Boundary Detection

This section describes the process of extracting road boundary points after acquiring point cloud data, as shown in Figure 3. First, the surface of the road surface is detected after the point cloud data are preprocessed. Then, after filtering the ground, the beam area is divided in the elevated point and the road boundary candidate points in the beam area are extracted. Finally, the candidate points are filtered to obtain road boundary points.

### 3.1. Data Preprocessing

The Velodyne HDL-32E LiDAR was installed on the mining truck. At the same time, the mining truck was also equipped with a GPS (Global Positioning System)/IMU (Inertial Measurement Unit) navigation and positioning system. The Velodyne HDL-32E LiDAR sensor has up to 32 lasers in a 40-degree vertical field of view. The sensor has a vertical field of view of +10${}^{\circ }$ to −30${}^{\circ }$ and provides a 360 degree horizontal field of view. The Velodyne HDL-32E produces a point cloud of 700,000 points per second with a scan range of 70 m and a typical accuracy of 2 cm. The LiDAR scan range is 360 degrees in the vertical field of view. The size of the mining truck itself affects the collection of point cloud data. To better detect road data ahead, the location of the sensor is very important. As shown in Figure 4, the sensor horizontal scan angle is 180 degrees in this mounting position.

The cartesian coordinate system is shown in Figure 4. The vertical ground is the *z*-axis and the truck front is the *x*-axis. Determine the direction of the *y*-axis coordinate system according to the right-hand coordinate system. To facilitate the processing of the point cloud data and the calculation of the vehicle position, the point cloud data are converted from the sensor itself as the coordinate system origin to the center of the vehicle rear axle as the coordinate system origin. As shown in Figure 5, the coordinates of point *P* in the $O_{xyz}$ coordinate system are P(x,y,z). The coordinates of the same point $P^{\prime }$ in the $O_{x^{\prime }y^{\prime }z^{\prime }}^{\prime }$ coordinate system are $P^{\prime }\left(x^{\prime }y^{\prime }z^{\prime }\right)$. The three angles and the three translations in the known coordinate system are obtained, and the coordinate transformation relationship of the two coordinate systems would be found to achieve coordinate transformation.

Since the Velodyne HDL-32E LiDAR is moving with the mining truck during one scan period of the LiDAR, this will result in distortion of the data acquired by the LiDAR. Distortion point cloud data show that each scan line is not a closed circle, but a broken circle. The change in the data position of the point cloud due to the movement of the vehicle affects the boundary extraction of the road. To solve this problem, we use the method of Wang G et al. [6] to process the original point cloud data. This method obtains the posture and position of the mining truck at a certain moment by the position and posture between two adjacent frames. It is assumed that the LiDAR scan period *C* is very short, and the position and attitude of the vehicle change linearly during the scan period *C*. The end horizontal angle of the current frame of the scan line is $\varphi_{1}$, and the rotation angle of the point $P_{i}$ is $\varphi_{2}$. The time required for LiDAR to rotate from the $P_{i}$ point to the end position is:(1)$$t_{i}=C*\left(\varphi_{1}-\varphi_{2}\right)/2\pi$$
The position and posture of the mining truck in each frame is obtained through a GPS/IMU navigation system. The displacement matrix $T_{i}$ and pose matrix $R_{i}$ of $P_{i}$ are calculated by linear interpolation. After distortion correction, the formula for the obtained $P_{i}^{\prime }$ point is as follows:(2)$$P_{i}^{\prime }=R_{i}P_{i}+T_{i}$$

### 3.2. Ground Surface Detection

Existing ground detection technology is mostly based on flat road [25,26,27]. In addition, methods for detecting uphill and downhill slopes are also available. Asvadi et al. [28] proposed a multi-plane detection method and a plane splicing method. Because the method can fit two planes in the same plane detection area at the turn of the mine road, it is not applicable to the mine. A double meshing method can effectively detect the ground information of uphill and downhill and the ground information of the turning. After the point cloud is divided, the height information between grids can be directly processed [13]. The ground detection method uses the double meshing method we proposed in another paper [29]. This algorithm is used to determine whether the point cloud belongs to the ground according to the conditions set by adjacent grids. The object for the first grid division is the original point cloud data. Each grid has three values, and the grid size is 20 cm × 20 cm. The three values are the highest value $Z_{max}$ and lowest value $Z_{min}$ in the point cloud in the grid, and the maximum height difference $Z_{dist}$ of the point cloud. After the first ground detection, some continuous raised obstacles may also be detected as ground. Therefore, a sliding window is used to optimize the ground point cloud. The object of the second grid division is ground point cloud data. Initially, the window head is the grid window to be detected, and the sliding step is a grid. The difference between the starting window and the rest of the windows is used to update the ground point cloud. Figure 6 shows the results of the ground surface of the mine road before and after the detection.

### 3.3. Candidate Points Extraction

The point cloud of the ground detection is filtered to obtain the elevated point cloud. Extracting road boundary points in the divided beam regions facilitates the detection of curves or intersections [10]. To better detect various types of road conditions of the mine road, the boundary points are extracted in the beam region. The location of the emission point in the beam model directly affects the detection results. The location selection of the launch point may cause some road point cloud information to be occluded. We use the center point of the extracted road boundary points as the emission points of the beam model. The filtered elevated point cloud data is segmented using a beam model. Each point P(x,y,z) is divided into a corresponding one of the beam regions. The area division formula is as follows:(3)$$Q_{i}=144*\frac{arctan\left(\frac{y-Y}{x-X}\right)+\pi }{2\pi },\left\{-25\leq x\leq 25,-10\leq y\leq 50\right\}$$
where x,y are the coordinates of the point cloud within a certain range of P(x,y,z). $Q_{i}$ is the *i*th beam area, with a total of 720 beam areas. X,Y are the coordinate positions of the launch point and all the beams intersect at this point. As shown in Figure 7, the point cloud above the ground is divided into different beam regions. After the point cloud is divided into beam models, a point closest to the emission point is extracted as a road boundary candidate point in each beam region.

### 3.4. Boundary Points Fitting

As shown in Figure 8a, there are many false points in the directly extracted road boundary candidate points. These false road candidate points are basically caused by weeds, trees, and stones outside the road. We propose a series of methods including point cloud classification, distance filter, and RANSAC (RANdom SAmple Consensus) to fit the boundary points. The specific process is described below. First, we classify the extracted point clouds, as shown in Figure 8b. The candidate points extracted follow the direction angle. According to the point cloud save order, if the point cloud meets the following conditions, it is the same class.
(4)$$dist_{i}=\sqrt{{\left(x_{i}-X\right)}^{2}+{\left(y_{i}-Y\right)}^{2}}$$
(5)$$\left|dist_{1}-dist_{2}\right|>T_{a}$$
(6)$$ang_{i}=arctan\left(\frac{y_{i}-Y}{x_{i}-X}\right)$$
(7)$$\left|ang_{1}-ang_{2}\right|>T_{b}$$
where $dist_{i}$ is the distance between the candidate point and the emission point (X,Y) and $ang_{i}$ is the difference in angle between the candidate point and the emission point (X,Y). If the distance difference between the adjacent candidate points meet the threshold $T_{a}$ and the angle difference between the adjacent candidate points meet the threshold $T_{b}$, the points are classified into the same category. Then, in each point cloud class, the false candidate points are eliminated by using the distance filter, as shown in Figure 8c. The following equation represents the distance filter applied:(8)$$f_{dist}=n\left|y_{0}-y_{1}\right|+m\left|y_{0}-y_{2}\right|$$
where n and m are the coefficients of the formula. Among them, after the angle sorting, $y_{1}$ is a point before $y_{0}$ and $y_{2}$ is a point after $y_{0}$. Three adjacent points are judged based on the positional information in the horizontal direction. The previous point has a larger impact on the actual detection of the road, thus the value of *n* is greater than *m*. Finally, the candidate points are fitted and filtered using the RANSAC algorithm. The least squares fitting is to fit all points including noise points [6]. RANSAC estimated polynomial model avoids noise fitting through multiple iterations. After obtaining the final extracted boundary points, the boundary point fitting is performed using the polynomial model estimated by the RANSAC algorithm, as shown in the Figure 9.

## 4. Road Boundary Tracking

This section introduces the method of boundary point tracking after boundary point detection, as shown in Figure 3. Road tracking uses predicted values to avoid false and missed detection of boundary points. We use KF (Kalman Filter) to predict road boundary points. We use a correlation method based on the direction of travel of the vehicle to associate the predicted boundary points with the detected boundary points.

Kim Z et al. [30] used particle filtering to track the curb due to the predicted unreliability of vehicle motion. Zhang Y et al. [31] built a prediction model for road boundary points and used Kalman Filter for prediction and updating. However, this method does not use the results of road boundary point tracking to improve the results of road detection. In [6], KF is also used for road curb tracking. The method uses an Amplitude-Limiting filter to determine whether the predicted point and the current difference satisfy the threshold. If this threshold is not met, the predicted value is used to replace the current one. The method tracks three points on the left and right boundaries of the road. The small number of tracking points makes it impossible to track curves based on these points. Therefore, this method does not apply to the boundaries of the road at turns. The prediction result of the previous frame boundary point is associated with the result data of the current frame in the geodetic coordinate system. For multi-target tracking data association algorithms, the measured and predicted values are the same target. Because the boundary points of different frames are not the same target, algorithms such as PDA (Probabilistic Data Association) [32] or JPDA (Joint Probabilistic Data Association) [33] cannot be used. In this paper, an algorithm for the association of boundary points is proposed, which matches the boundary points of two adjacent frames. Finally, the current road boundary point detection result is updated according to the predicted point.

### 4.1. Road Boundary Point Prediction

As shown in Figure 10, the three coordinate systems are the k time mining truck body coordinate system, the *k* + 1 time truck body coordinate system, and the geodetic coordinate system. Road boundary point detection is based on the vehicle body coordinate system. Before the boundary point tracking algorithm, the vehicle body coordinate system is converted to the geodetic coordinate system under the GPS/IMU system. According to the curb prediction model proposed in [32], the results of the state matrix and error covariance matrix at prediction time *k* + 1 based on time *k* are as follows:(9)$$\hat{x}\left(k+1\right)=A\left(k\right)X\left(k\right)+B\left(k\right)\mu \left(k\right)+\omega \left(k\right)$$
(10)$$\hat{P}\left(k+1\right)=A\left(k\right)P\left(k\right)A^{T}\left(k\right)+Q$$
where
(11)$$X\left(k\right)=\left[\begin{array}{c} x(k)\\ y(k) \end{array}\right]\mu \left(k\right)=\left[\begin{array}{c} \Delta x(k)\\ \Delta y(k) \end{array}\right]$$
(12)$$A\left(k\right)=\left[\begin{array}{cc} cos(\phi_{k}) & sin(\phi_{k})\\ -sin(\phi_{k}) & cos(\phi_{k}) \end{array}\right]$$
(13)$$B\left(k\right)=\left[\begin{array}{cc} -cos(\phi_{k}) & -sin(\phi_{k})\\ sin(\phi_{k}) & -cos(\phi_{k}) \end{array}\right]$$

x(k) and y(k) are in the set of left boundary points and road right boundary points of mine road. A(k) and B(k) are state transition matrices. Δx(k) and Δy(k) are moving distance between two frames of *x*-axis and *y*-axis. $\phi_{k}$ is the rotation of the vehicle. X(k+1) is the predicted next frame state and x(k) is the estimation of the current state. $\hat{P}\left(k+1\right)$ is the error covariance prediction of the next state and P(k) is the error covariance estimation of the current state. ω(k) is white Gaussian noise with covariance matrix *Q*. As we have detected the road boundary points, we assume that the measurement noise is white Gaussian noise.The measurement vector is represented by:(14)z(k)=x(k)+ν(k)
where ν(k) is the measurement noise with a covariance matrix *S*. The process of status update is as follows:(15)$$G_{k}\left(k+1\right)=\hat{P}\left(k+1\right)*{\left[\hat{P}\left(k+1\right)+S\right]}^{-1}$$
(16)$$X\left(k+1\right)=\hat{X}\left(k+1\right)+G_{k}\left(k+1\right)\left(z\left(k+1\right)-\hat{X}\left(k+1\right)\right)$$
(17)$$P\left(k+1\right)=\hat{P}\left(k+1\right)-G_{k}\left(k+1\right)\hat{P}\left(k+1\right)$$
where $G_{k}\left(k+1\right)$ is the the Kalman filter gain, *S* is the measurement noise covariance, X(k+1) is the final output of the tracking algorithm which represents one road boundary point, and P(k+1) represents the uncertainty of the current state estimate. The boundary points on both sides of the road are predicted according to the above formula.

### 4.2. Road Boundary Points Association

In this paper, two different road boundary points of adjacent frames are associated. The road boundary points are associated in a direction perpendicular to the direction of advancement of the mining truck. The body of the mining truck is the origin of the coordinate system; the direction expression is as follows:(18)$$y-\frac{x}{\tan(90-\varphi )}-b=0$$
where φ is the yaw angle of the vehicle obtained according to the GPS/IMU system. *x* and *y* are the horizontal coordinate and vertical coordinate of the autonomous truck. *b* is the interval between the road boundary point and the point in the direction of travel of the vehicle. In this paper, the road boundary point associated operation is performed every 20 cm. The predicted point of the previous frame is centered on the current detection point in the direction of the above expression. The boundary point associated with the circle is within 20 cm. As shown in Figure 11, the green circle is the range in which the road boundary points are determined. The green gate is in the direction of the above expression. If the closest point is obtained within the range, the road boundary point is successfully associated. Otherwise, the road boundary point association will fail.

### 4.3. Road Boundary Points Update

The detection of road boundary points is actually not smooth enough, thus not all road boundary points in this paper use the value of the detection algorithm to update the tracking results. The current road boundary point is considered valid only when the current detection result satisfies the following requirements, and is input to the Kalman Filter at the updating step. Otherwise, the prediction of the previous frame of the associated data is used as the current detection result. The threshold requirements are shown in the following formula:(19)$$\left|X_{pre}-X_{cur}\right|\leq R_{a}$$
(20)$$\left|Y_{pre}-Y_{cur}\right|\leq R_{b}$$
where $X_{pre}$ and $Y_{pre}$, respectively, represent the horizontal and vertical coordinates of the road boundary points detected in the previous frame. $X_{cur}$ and $Y_{cur}$, respectively, represent the horizontal and vertical coordinates of the road boundary points detected in the previous frame. If the above expressions meet the thresholds $R_{a}$ and $R_{b}$, the boundary point detection value is used to update the result of the final Kalman Filter. The role of tracking road boundary points is to stabilize and improve road detection results. If the road boundary prediction point is not successfully associated with the detection result point, the tracked prediction point is used as the current detection result. If the boundary points does not satisfy the above threshold condition after successful association, the predicted point is used as the result of the current detection.

## 5. Experiment Results

This section describes the experimental setup, evaluation of the road detection algorithm, and detection results for autonomous truck. The experimental setup describes the application environment of the algorithm. In addition, the scene of data collection and the data annotation situation are explained. The evaluation of the algorithm compared the detection points with the real points on straight roads and curved roads. At the end of this section, the detection results of autonomous trucks and the handling of missed detection on road boundary points and false detection problems are shown.

### 5.1. Experimental Setup

The Velodyne HDL-32E sensor was installed in front of the mining truck and collected data in the mine scene. The LiDAR sensor was installed on mining trucks at a height of 3.61 m. The speed of the truck during the data collection was 20–25 km/h. The sensor dataset collected based on the Linux operating system was 0.1 s per one frame point cloud data. The point cloud dataset was collected for algorithm improvement under the ROS system. The algorithm was implemented by C++ and PCL on Ubuntu 16.04. The algorithm was validated offline in the dataset and then actually verified on an automated driving mining truck. After the detection algorithm obtained a high accuracy rate, the code was uploaded to the IPC (Industrial Personal Computer) of the mining truck to detect road in real time.

In Inner Mongolia, China, point cloud data of mining environment were collected by autonomous truck. The scenes of the collected data included straight roads, left turn roads, right turn roads, uphill roads, and downhill roads in the mine. Figure 12 shows the real scene of the mine dataset. Due to the slow speed of mining trucks, the total data collected on different road sections are 30 min. The total length of the mine roads in the dataset is approximately 11 km. First, the collected raw data were parsed into a PCD format. A PCD file is a frame of point cloud data collected by LiDAR, thus the similarity of scenes in each frame file is very high. Then, we selected 7000 PCD files from the parsed 18,000 PCD files for data annotation. Finally, we marked the road boundary points and ground points of the original point cloud data. Figure 13 shows the interface of the point cloud data annotation tool. This annotation tool can mark obstacles, lane lines, ground, and road boundaries. The index file generated after labeling contains the location information of each point belonging to the road boundary. In the mine dataset, the accuracy of the algorithm was verified based on the road boundaries marked by the data.

### 5.2. Evaluation of the Road Detection Algorithm

We evaluated the algorithm on our mine dataset labeled with road boundary points. The accuracy of road boundary point detection is determined by the effective NN (nearest neighbor). Because of the positioning error and the marking error, the detected road boundary point and the marked road boundary point distance threshold were set to 8 cm. A point smaller than the distance threshold was considered to be the correct detection result. After verifying the original data by the algorithm, the position of the road boundary point was compared with the position of the marked boundary point. We introduce three indicators for evaluation.

Precision denotes the fraction of the road boundary points detected correctly out of all the road boundary points detected in one frame:
$$Precision=\frac{TP}{TP+FP}$$
where TP is the true positive numbers and FP is the false positive numbers (false alarm).Recall denotes the fraction of the road boundary points detected correctly out of all the labeled road boundary points:
$$Recall=\frac{TP}{TP+FN}$$
where FN means false negative (missed detection).$F_{1}$ denotes the harmonic average of precision and recall:
$$F1=2*\frac{Precision*Recall}{Precision+Recall}$$

There are few existing technologies and studies on unstructured mine road detection. Han J et al. [9] obtained boundary points on each scanning line to detect unstructured country roads. Han J et al. [9] extracted road boundary points based on the resolution of the LiDAR and information between the point cloud. The method calculates the distance and angle between the point cloud and the origin on each scan line. Then, the difference between adjacent point clouds is calculated according to the obtained angle and the obtained distance. In this paper, this method is used as the verification of mine road detection by Algorithm 1. It can be seen from the results that the method of road boundary detection based on scan lines is currently not applicable to mine roads. Wang et al. [5] filtered the ground to extract the boundary points of the road at the elevated point. Since the method of Wang et al. [5] is applied to urban roads, it is not applicable to mine roads. Algorithm 2 replaces the method of urban ground detection in [5] with the ground detection algorithm proposed in this paper.

The results of different contrast methods for detecting boundary points on both sides of the road are recorded in MATLAB. In Figure 14 and Figure 15, the three algorithms are compared on a straight road and a curved road. The blue mark is the detected road boundary point and the red mark is the actual boundary point on the mine road. Algorithm 1 cannot accurately extract the road boundary points because the difference in characteristics between the points on the ground and the boundary points on the scan line is complicated. The result of Algorithm 2 is better than that of Algorithm 1, but the accuracy of the boundary point extraction results is not good. The results above show that our proposed method for detecting road boundary points is superior to other methods. Table 1 shows the evaluation results of the three comparison methods on different roads. In this paper, roads that change the direction of the vehicle by more than 60 degrees are considered as curved roads, thus the rest are regarded as straight roads. The results of straight roads in these three methods are better than the results of the curve roads. The reason for this result is that there may be a case where the height of the road boundary point is small or has no boundary at the curve of the mine road. The number of boundary points existing in actual mine roads is very small, thus the recall rate is generally low. The method proposed in this paper has a boundary point detection rate of 90%, which can accurately detect mine roads.

An autonomous truck can accurately detect road boundary to ensure driving safety. After the autonomous truck obtains the position of its own vehicle through the GPS/IMU navigation system, the perception system calculates the distance between the own vehicle and the boundary points on both sides of the road. Road boundary detection helps autonomous trucks maintain a safe distance from road boundary to avoid accidents.

### 5.3. Detection Results for Autonomous Truck

The proposed algorithm was tested on the autonomous truck platform. The IPC (Industrial Personal Computer) was used on the autonomous truck to verify the uploaded algorithm in real time. The Nuvo-5100VTC is a high-performance processor embedded controller with an Intel core i7-6700 processor. This Industrial Personal Computer Cache is 8 M, with a base frequency of 2.3 Ghz, which can process point cloud data in real time. The truck speed during the actual inspection of the mining truck is 23 km/h. The result of real-time road detection are displayed by RVIZ in ROS system. As shown in Figure 16, the blue point is the original point cloud data of the mine road. The red mark is the result of fitting the road boundary line of the mine road. The results of straight roads and turning roads show that the proposed method is accurate and effective. In the uphill and downhill road scenes, the results of road detection are not as good as the results of the flat road. For large-slope roads, the point cloud at 14 m in front of the mining truck is already very sparse. Due to the lack of point cloud data, the boundary line fitting results of the slope road are not as good as the results of the flat road detection. After extracting the road boundary points, the RANSAC algorithm is used to filter the boundary points while fitting the road boundary curves. The advantage of the RANSAC algorithm is that it can actively filter out points outside the road boundary model.

Due to the limitations of the algorithm, there are false detection situations in the road boundary point detection process. In addition, road boundary point miss detection is caused by the instability of LiDAR data acquisition during vehicle travel. Therefore, the road boundary point tracking algorithm can solve the above problem. If the predicted road boundary point is not associated with the detection result, the predicted value is used instead of the detected value. If there is a large deviation between the detection point and the tracking point of the previous frame, the predicted value of the previous frame is used as the detection result of the current frame. In this way, the prediction of road boundary points optimizes the real-time detection results. This paper successfully solves the difficult problems of mine road detection and realizes the effectiveness of road detection. Experiments show that this method can effectively detect mine roads and real-time road boundary detection can be achieve on these datasets.

## 6. Conclusions

This experiment verified and evaluated the unstructured mine road detection method proposed in this paper. This paper can effectively realize ground detection for the characteristics of bumps or depressions on mine road pavement. After filtering the road surface, the boundary point is extracted from the beam model at the elevated point. In this paper, several operations such as distance filter, boundary point classification, and RANSAC curve model filtering are used to optimize the boundary points. This paper solves the problem of missed detection and false detection by tracking the boundary points. Experiments showed that the method can be applied to the characteristics of mine roads and has better road detection accuracy and robustness. At present, for the study of unstructured road detection, it is a great challenge to detect mine roads based on the characteristics of scanning lines. Detection of boundary points on elevated points is more suitable for mine roads. The method proposed in this paper was in experiments used after obstacles such as vehicles traveling on the road surface were removed. Further work will focus on detecting drivable areas when there are obstacles on the road. Finally, the road surface obstacles and road boundaries are simultaneously detected. L-shaped road boundary line fitting is also the focus and difficulty of future work research.

## Figures and Tables

**Figure 1 sensors-20-01121-f001:**
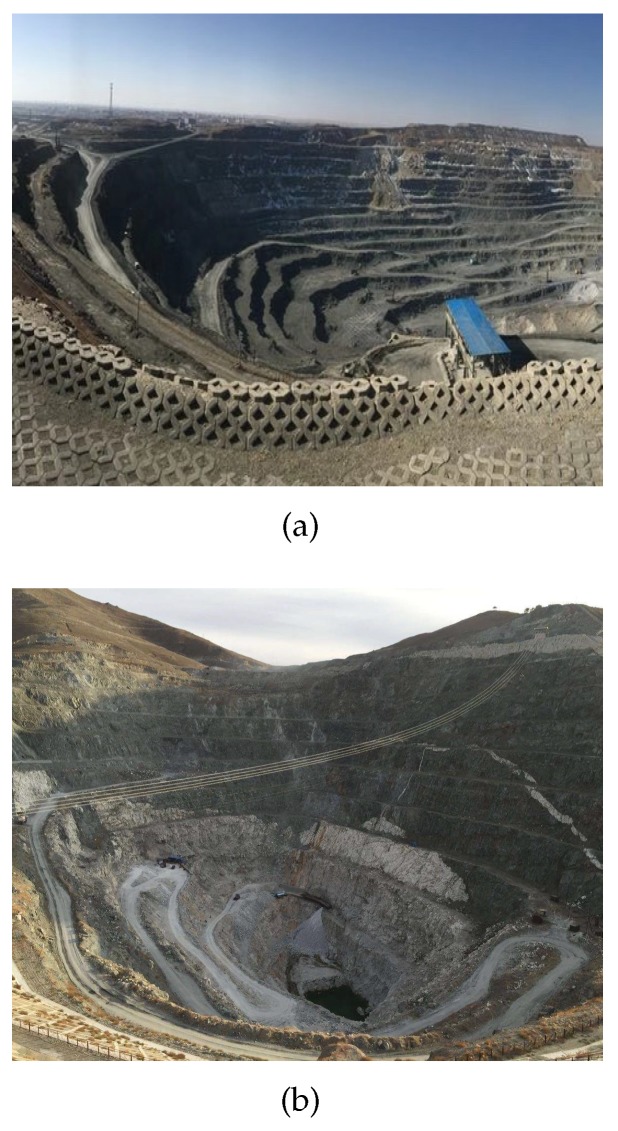
Mine actual environment.

**Figure 2 sensors-20-01121-f002:**
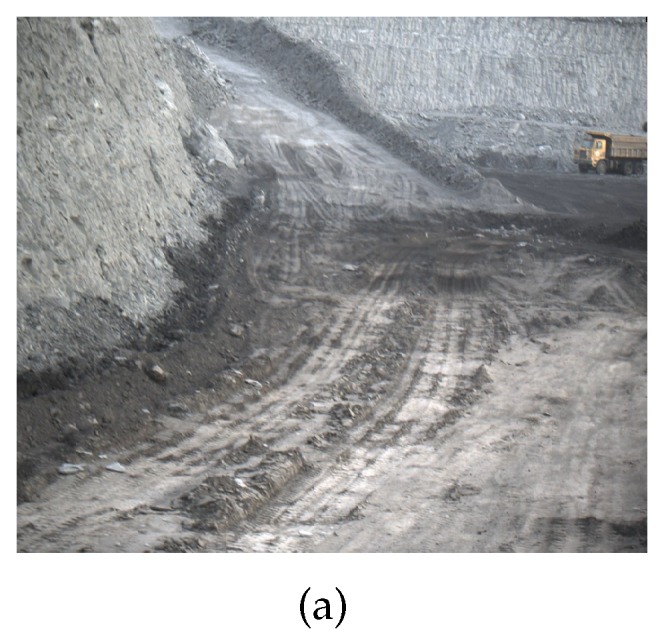
(**a**,**b**) are mine road; and (**c**) is the original point cloud data.

**Figure 3 sensors-20-01121-f003:**
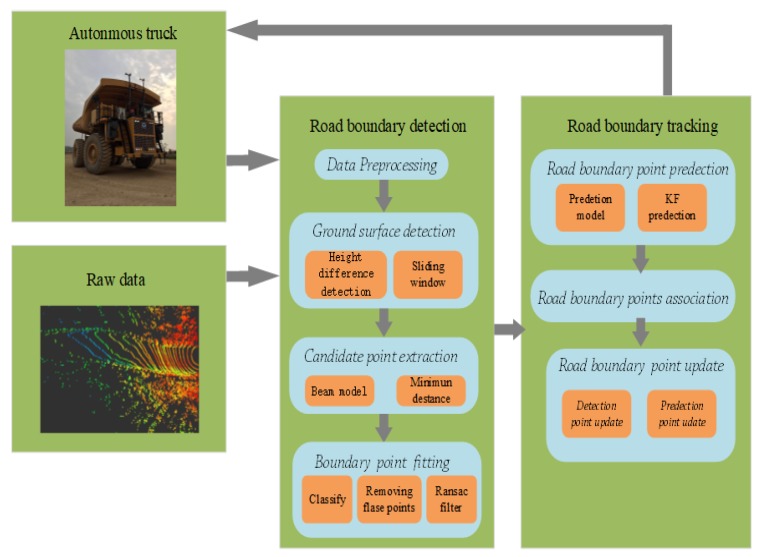
Mine road detection structure.

**Figure 4 sensors-20-01121-f004:**
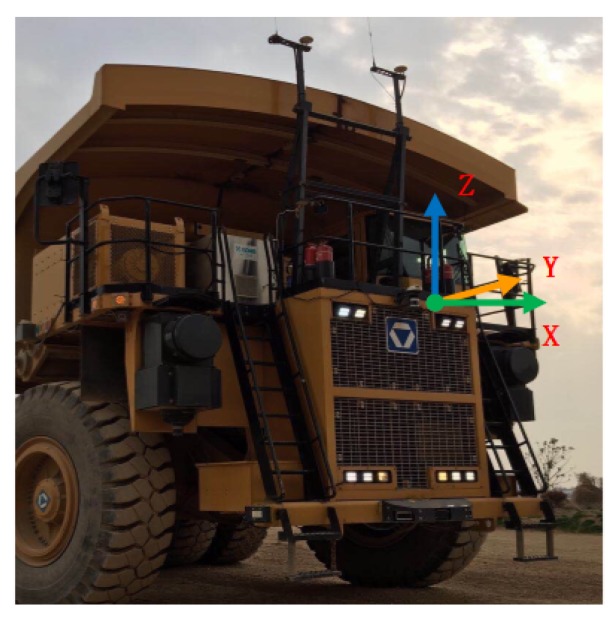
The cartesian coordinate system of LiDAR.

**Figure 5 sensors-20-01121-f005:**
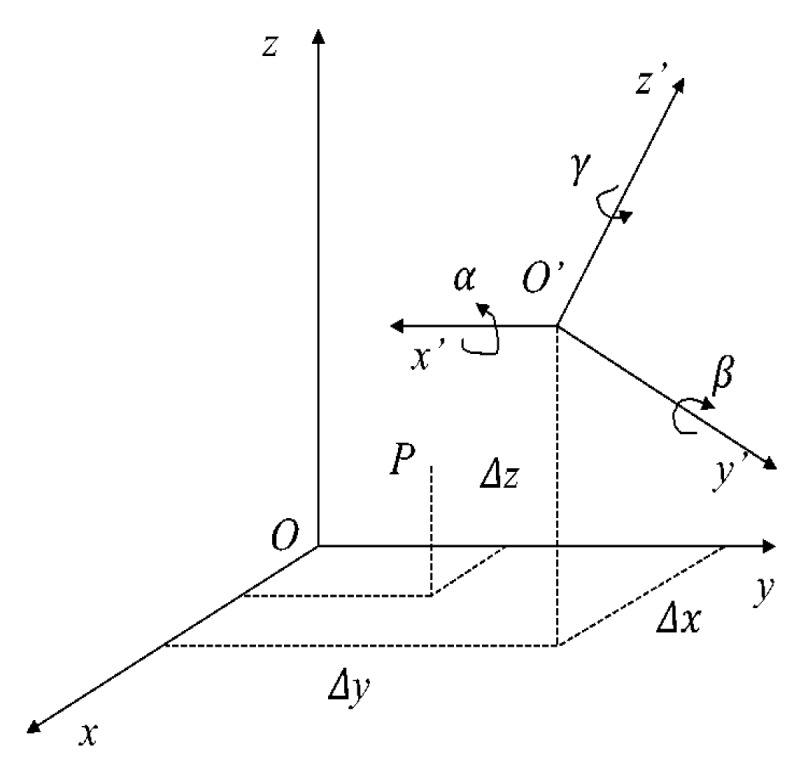
Conversion between two coordinate systems.

**Figure 6 sensors-20-01121-f006:**
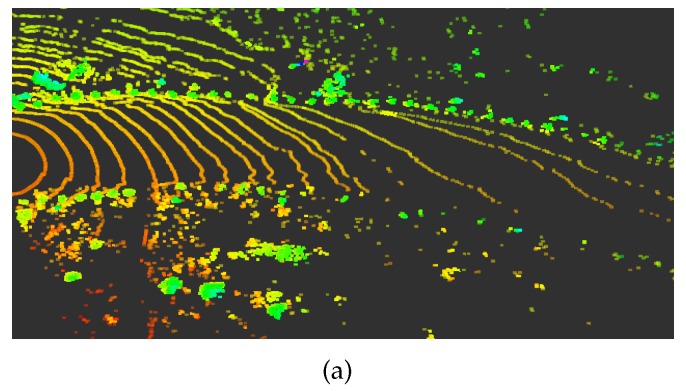
(**a**) The data before the ground detection; and (**b**) the elevated point data after the ground is removed.

**Figure 7 sensors-20-01121-f007:**
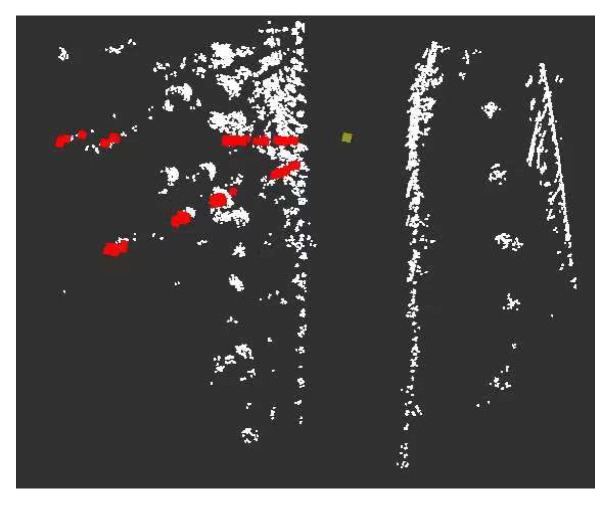
Beam model division. Yellow represents the emission point of the beam model. Red represents the point clouds in the 40th and 50th beam regions.

**Figure 8 sensors-20-01121-f008:**
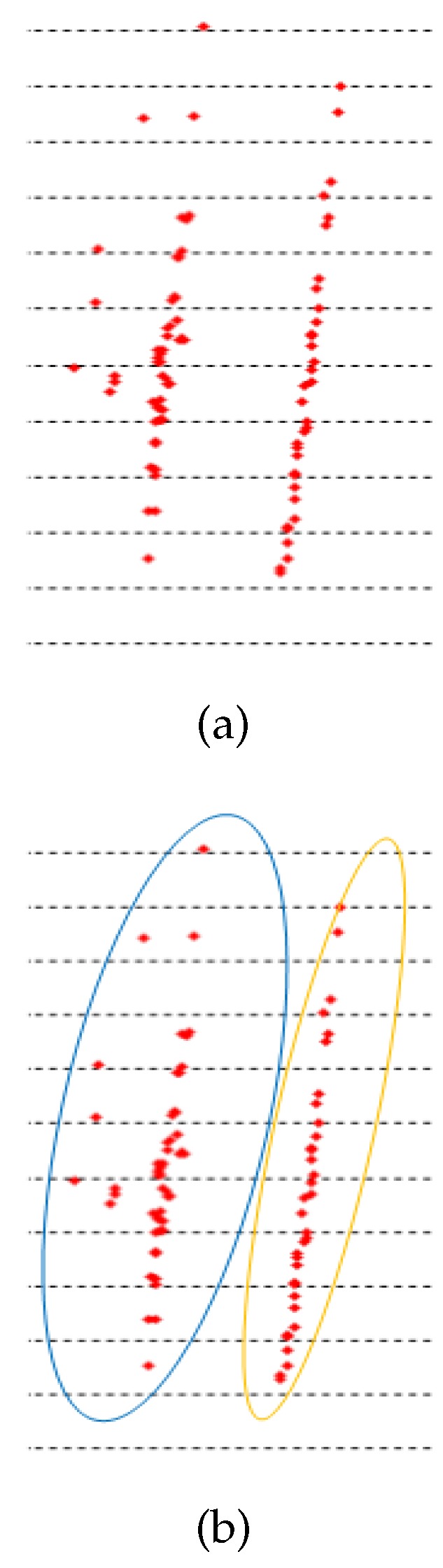
(**a**) The candidate point result; (**b**) the classified candidate point result; and (**c**) the candidate point filtered result

**Figure 9 sensors-20-01121-f009:**
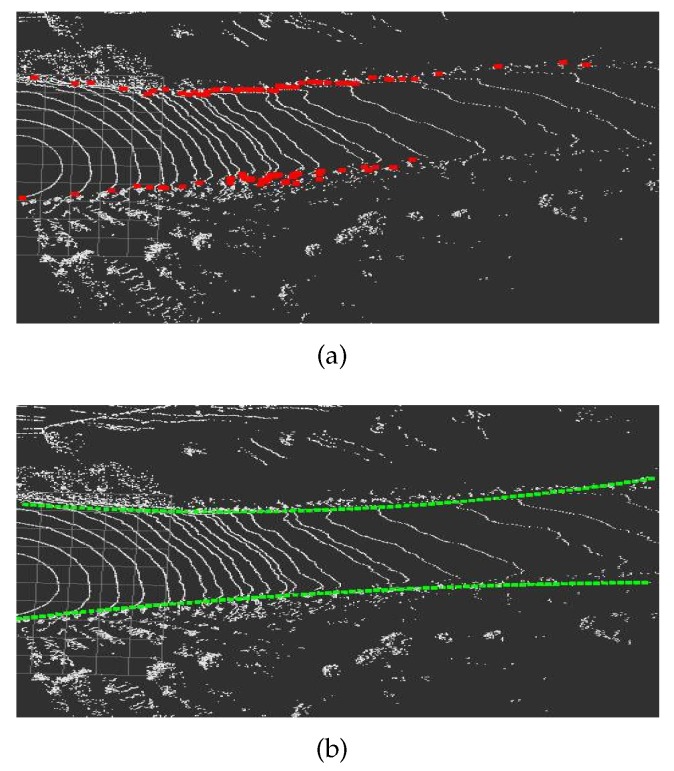
(**a**) The road boundary point extraction result; and (**b**) the road boundary line fitting result.

**Figure 10 sensors-20-01121-f010:**
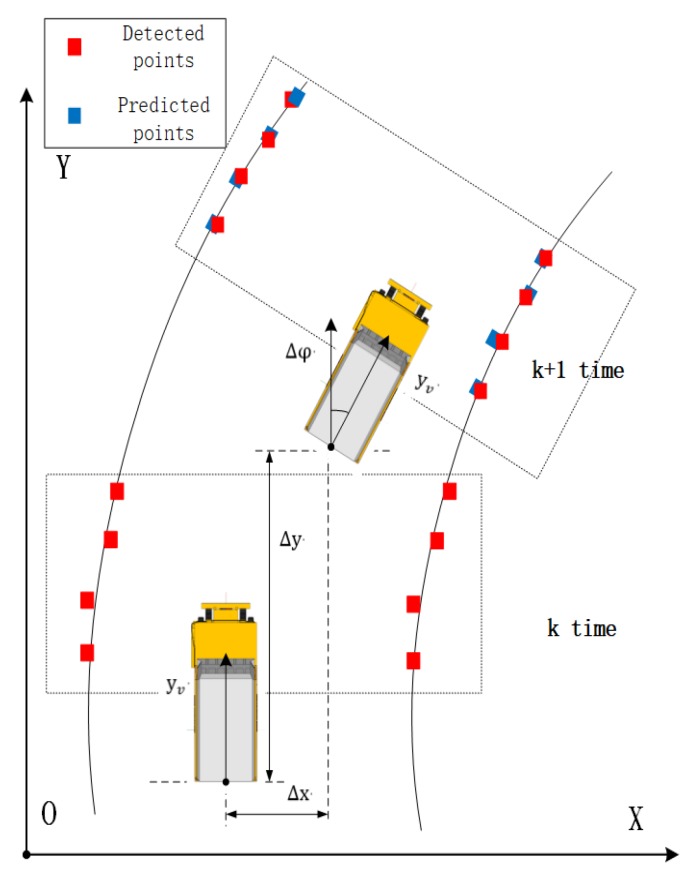
Coordinate frames of road points tracking algorithm.

**Figure 11 sensors-20-01121-f011:**
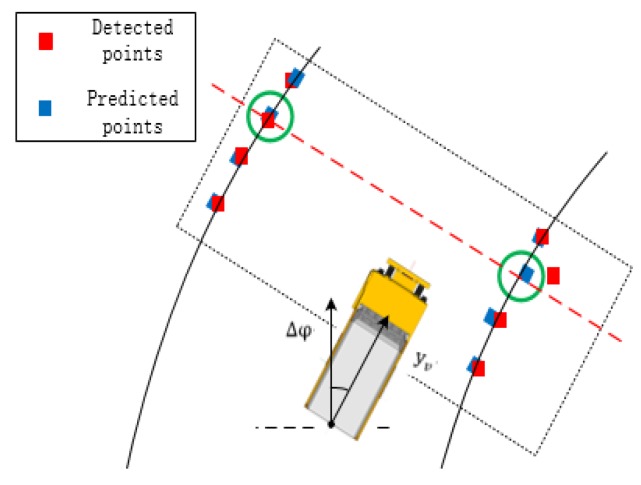
The gate of road boundary point association.

**Figure 12 sensors-20-01121-f012:**
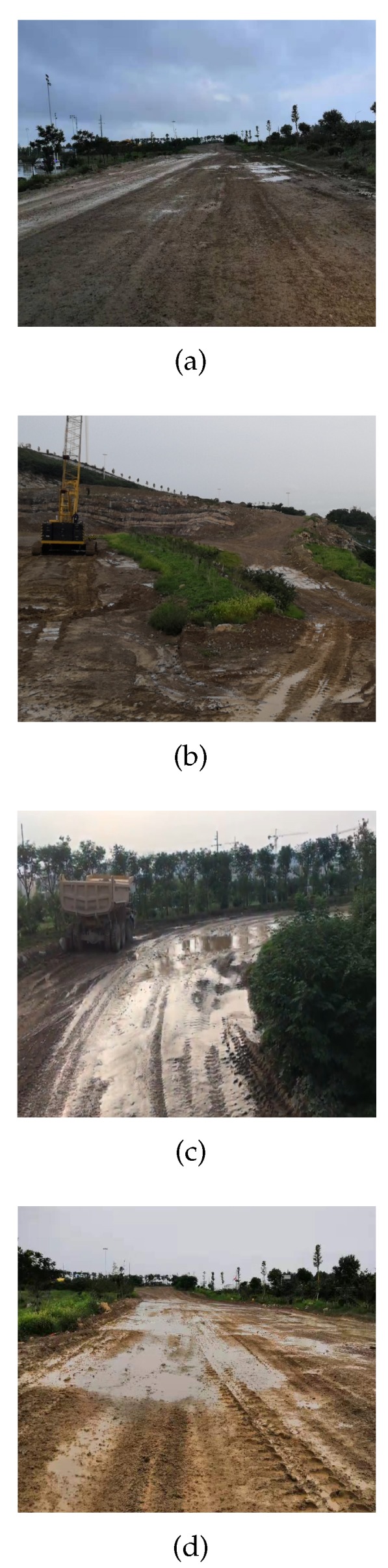
The real scene of the mine dataset.

**Figure 13 sensors-20-01121-f013:**
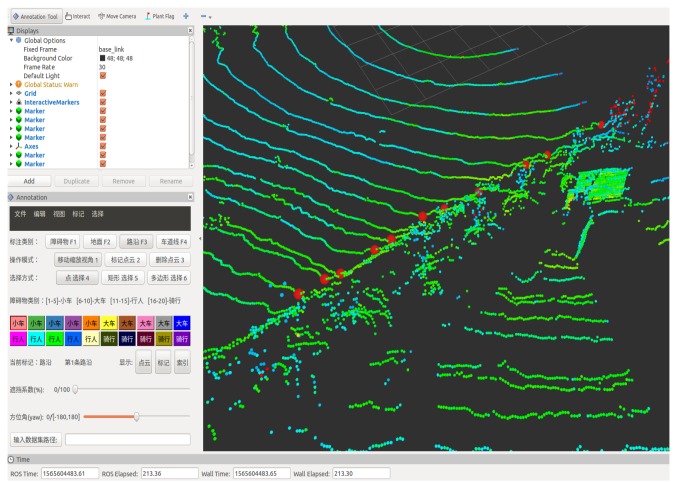
Point cloud data annotation tool interface.

**Figure 14 sensors-20-01121-f014:**
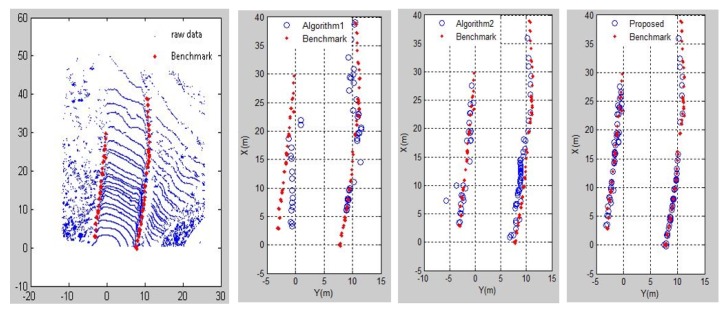
The results of three methods on straight road scenario.

**Figure 15 sensors-20-01121-f015:**
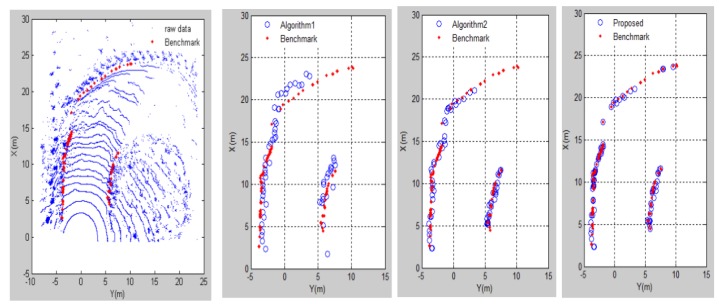
The results of three methods on curved road scenario.

**Figure 16 sensors-20-01121-f016:**
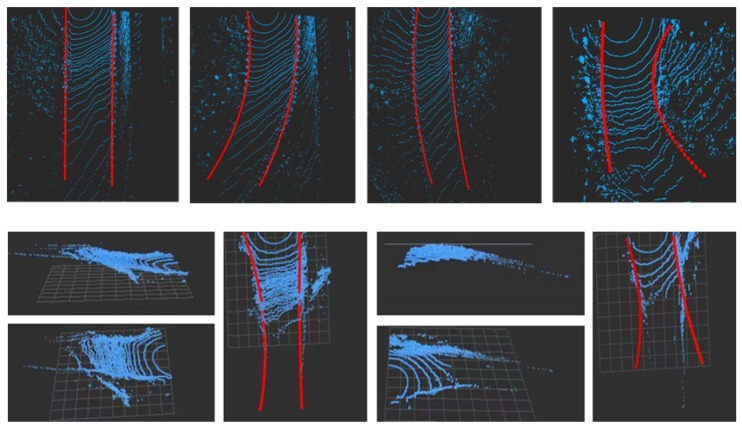
The real-time results of some road scenes.

**Table 1 sensors-20-01121-t001:** Evaluation results of road boundary detection algorithm.

	Methods	Straight Road	Curved Road
Precision	Algorithm 1	75.44%	71.88%
	Algorithm 2	87.93%	86.89%
	Proposal	93.65%	91.14%
Recall	Algorithm 1	54.32%	46.48%
	Algorithm 2	62.96%	53.54%
	Proposal	72.84%	77.78%
F1	Algorithm 1	63.16%	56.45%
	Algorithm 2	73.38%	66.26%
	Proposal	81.94%	83.932%

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
