# Peer review of "Real-Time Mine Road Boundary Detection and Tracking for Autonomous Truck"

_sensors, 2020, doi:10.3390/s20041121_

Round 1

Reviewer 1 Report

This is a highly relevant paper about a hot topic. Benefits of self-managed road traffic will - once it has become the standard for operating all vehicles in - be used on a global scale and as such work for e.g. smaller energy consumption, less accidents and more efficient traffic control.

Can't think of anything to be done in a different way re: the manusript.

Author Response

Thank you very much for reviewing this paper. I have improved my paper better.

Reviewer 2 Report

This paper presents a novel method to detect the boundaries of mine roads for autonomous truck driving in the mining environment.

The paper contains a comprehensive introduction and literature review that frames the problem into a well defined context. The solution presented is novel, and its empirical comparison to two other existing methods yields promising results.

The paper would benefit from the following improvements:

The title should be made more specific to mine roads, because there is no evidence presented that the method can be applied to other environments. In the experimental section, you decide to use Precision, Recall, and their harmonic mean to measure the effectiveness of your algorithm. It would help the paper to explain why these measures are appropriate. It would be useful to determine what effect the improvement in these measures brings to the actual task of autonomous driving. Is the truck able to make faster decisions? Is it safer? Is it more controllable? Is is less likely to get stuck or to plunge back down into the mine? More extensive editing is needed for clarity and readability. In line 62 the paper is referred to as a "thesis". This should be changed.

With these changes, the paper would be an excellent candidate for publication.

Author Response

Q1: The title should be made more specific to mine roads, because there is no evidence presented that the method can be applied to other environments.
A1: Title has been renamed:Real-time mine road boundary detection and tracking for autonomous truck.
  Q2: In the experimental section, you decide to use Precision, Recall, and their harmonic mean to measure the effectiveness of your algorithm. It would help the paper to explain why these measures are appropriate. It would be useful to determine what effect the improvement in these measures brings to the actual task of autonomous driving. Is the truck able to make faster decisions? Is it safer? Is it more controllable? Is is less likely to get stuck or to plunge back down into the mine?
A2: In the experimental section,we have explained how road boundary line detection can improve autonomous truck driving.
  Q3: More extensive editing is needed for clarity and readability.
A4: Expressions throughout the paper have been appropriately modified.
  Q4: In line 62 the paper is referred to as a "thesis". This should be changed.
A4: It has been modified.

Reviewer 3 Report

This article describe the algorithms for detecting mine road boundary. Final results looks normal,  but there are many typing errors, undefined equations, and insufficient explanation  for the equation and texts. To publish in this article, it need more careful explanation and correct typing error.

For example

1.Equation (1) : What does C means?

2. Equation (3) , I cannot find index i in x, y,

 And for equation (3) and (4) we need figure. Figure is more easy to understand. Please include figure.

3.Equation (7), (9) no definition for T, R

4.Between Equation (10) (14) and text ?   ..(k) is rotation of the vehicle???

(k) is is white Gaussian noise .???

We cannot find the definition or explanation for the matrix A(k), B(k)…

5.”The displacement matrix Ti and the pose matrix Ti of the current” : The same Ti (Typing error?)

6.line 84: many false points. [21]use the ground :typing error makes misunderstanding

7.line 165 : “the ground detection method use Double….in other paper” : need reference

8.from line 159-174 : N*N, 5*N*N ??’ : need more careful explanation

9.and KF, RANSAC,SVM,…. : at first acronym add informations about these acronym.

10. line 226 : lter, nal ???

11. and others

Author Response

Thank you for your suggestions for this paper. We have carefully revised the paper as follows:

Q 1-5: Variables in the formula are not explained clearly.

A 1-5: We have modified and explained the relevant variables in the formulaFor question 2, we added pictures to explain the beam model.

Q6:line 84: many false points. [21]use the ground :typing error makes misunderstanding

A 6:We have modified this sentence.

Q7:line 165 : “the ground detection method use Double….in other paper” : need reference

A7:We have cited this reference.

Q8: from line 159-174 : N*N, 5*N*N ??’ : need more careful explanation

A8:We have explained it specifically in the paper.

Q9:and KF, RANSAC,SVM,…. : at first acronym add informations about these acronym.

A9: We have added information about these acronyms.

Q10:line 226 : lter, nal ???

A10: We have modified the sentence.

Q11: and others

A11: We have explained all the variables of the entire formula.